

# An Approach to Computing Discrete Adjoints for MPI-Parallelized Models Applied to the Ice Sheet System Model

Eric Larour[1] , Jean Utke[2] , Anton Bovin[3] , Mathieu Morlighem[4] , and
Gilberto Perez[5]

[1]Jet Propulsion Laboratory - California Institute of technology, 4800 Oak Grove Drive MS
300-323, Pasadena, CA 91109-8099, USA
[2]Allstate Insurance Company, 2775 Sanders Rd., Northbrook, IL 60062, USA.
[3]The University of Chicago, Chicago, Illinois, USA.
[4]University of California, Irvine, Department of Earth System Science, Croul Hall, Irvine, CA
92697-3100, USA
[5]University of California, Irvine, School of Information and Computer Sciences, Irvine, CA
92697-3100, USA

*Correspondence to:* Eric Larour (eric.larour@jpl.nasa.gov)

**Abstract.**

Within the framework of sea-level rise projections, there is a strong need for hindcast validation of the evolution of polar ice sheets in a way that tightly matches observational records (from radar, gravity, and altimetry observations mainly). However, the computational requirements for making
5  hindcast reconstructions possible are severe and rely mainly on the evaluation of the adjoint state of transient ice-flow models. Here, we look at the computation of adjoints in the context of the NASA/JPL/UCI Ice Sheet System Model, written in C++ and designed for parallel execution with MPI. We present the adaptations required in the way the software is designed and written but also generic adaptations in the tools facilitating the adjoint computations. We concentrate on the use of
10  operator overloading coupled with the AdjoinableMPI library to achieve the adjoint computation of ISSM. We present a comprehensive approach to 1) carry out type changing through ISSM, hence facilitating operator overloading, 2) bind to external solvers such as MUMPS and GSL-LU and 3) handle MPI-based parallelism to scale the capability. We demonstrate the success of the approach by computing sensitivities of hindcast metrics such as the misfit to observed records of surface altime-
15  try on the North-East Greenland Ice Stream, or the misfit to observed records of surface velocities on Upernavik Glacier, Central West Greenland. We also provide metrics for the scalability of the approach, and the expected performance. This approach has the potential of enabling a new generation of hindcast-validated projections that make full use of the wealth of datasets currently being collected, or alreay collected in Greenland and Antarctica.





## 1  Introduction

Constant monitoring of polar ice sheets through remote sensing, in particular since the advent of altimeter, radar, and gravity sensors such as ICESat-1, CryoSat, RADARSAT-1, ERS-1 and ERS-2, Envisat, and GRACE, has created a large amount of data that has yet to find its way through Ice Sheet Models (ISMs) and hindcast reconstructions of polar ice sheet evolution. In particular, as evidenced by the wide discrepancy between ISMs involved in the SeaRISE and Ice2Sea projects (Nowicki et al., 2013; Bindschadler et al., 2013) significant improvements in modeled projections of the mass balance of polar ice sheets and their contribution to future sea-level rise has not resulted from the increase in availability of data, but rather from improvements in the type of physics captured in forward models. One reason for this is the lack of data assimilation capabilities embedded in the current generation of state-of-the-art ISMs. In the past 10 years, great strides have been made in improving model initialization by using steady-state model inversions of basal friction (MacAyeal, 1993; Morlighem et al., 2010; Larour et al., 2012; Price et al., 2011; Arthern and Gudmundsson, 2010), ice rheology (Rommelaere and MacAyeal, 1997; Larour et al., 2005; Khazendar et al., 2007) and bedrock elevation (Morlighem et al., 2014) among others. However, these approaches aim at improving our knowledge of poorly constrained input parameters and boundary conditions as long as the ice-flow regime is captured in a steady-state configuration. These inversions rely on analytically derived adjoint states of forward stress-balance or mass-transport models, but do not extend to transient regimes of ice flow.

Applications to transient models and long temporal time series such as the ICESat/CryoSat continuous altimetry record from 2003 to present-day, have been much more rare, and to our knowledge, are limited to a few studies such as Heimbach and Bugnion (2009), Goldberg and Sergienko (2011), Goldberg and Heimbach (2013), Larour et al. (2014) and Goldberg et al. (2015) among others. The main issue here precluding widespread application of transient data assimilation lies in the difficulty of deriving temporal adjoints of transient models. Actually, in many cases, a manual derivation of the adjoint state of a forward model is not possible, especially where ice-flow physics are not differentiable. This is the case for example in thermal transient models, where the melting-point is a physical constraint (of the threshold type) that is imposed on temperature. This numerical issue can be mitigated by adopting different approaches, such as: 1) ensemble runs, as in Applegate et al. (2012), where model runs compatible with observations are selected; 2) methods similar to the flux-correction methods implemented in Aschwanden et al. (2013); Price et al. (2011) where boundary conditions are corrected in order to match time series of observations (tuning approach); 3) quasi-static approaches, where snapshot inversions are carried out in time, as in Habermann et al. (2012, 2013) and 4) sampling methods, which have the main drawback of being computationally very expensive (each sample at the cost of one forward run). Though this is not an exhaustive list of all available methods, the main advantage of adjoint driven inversions is that it relies on the exact sensitivity of a forward model to its inputs, hence ensuring a physically meaningful inversion.



Understanding sensitivities of a forward ice-flow model, which is needed to physically constrain a temporal inversion, requires computation of derivatives of model outputs to model inputs. If such derivatives are approximated by finite-difference schemes, they are subject to the tradeoff be-
tween approximation and truncation errors for the perturbation, which is aggravated for higher-order derivatives. If the derivatives are computed using algorithmic differentiation (AD) (Griewank and Walther, 2008), also known as automatic differentiation, then one can attain derivatives with machine precision provided the underlying program implementation of the numerical model is amenable to the application of an AD tool. In particular, this approach does not depend on the type of physics
relied upon, and it is transparent to the model equations, provided each step of the overall software is differentiable. Indeed, the AD method assumes a numerical model $M$ that is a function

$$
\begin{aligned}
\boldsymbol{f} : \mathbb{R}^n &\rightarrow \mathbb{R}^m \\
\boldsymbol{x} &\mapsto \boldsymbol{f}(\boldsymbol{x})
\end{aligned}
\tag{1}
$$

implemented as a computer program. The execution of the program implies the execution of a sequence of arithmetic operators such as $+, -, *$ and intrinsics $\sin, e^x$, and so forth to which the chain
rule can be applied. For each such elemental operation $r = \phi(a, b, \ldots)$ with result $r$ and arguments $a, b, \ldots$[1] we can write the total derivative as:

$$
\dot{r} = \frac{\partial \phi}{\partial a} \dot{a} + \frac{\partial \phi}{\partial b} \dot{b} + \ldots \quad .
\tag{2}
$$

For example, if $\phi$ is the multiplication operator $*$, i.e. $r = ab$ then one will get the product rule $\dot{r} = b\dot{a} + a\dot{b}$. Applying the above rule to each elemental operation in the sequence gives a method to compute:

$$
\dot{\boldsymbol{y}} = \boldsymbol{J}\dot{\boldsymbol{x}} \quad \text{with the Jacobian} \quad \boldsymbol{J} = \left[ \frac{\partial f_i}{\partial x_j} \right], i = 1 \ldots m, \, j = 1 \ldots n
$$

without explicitly forming the Jacobian. This method applies the chain rule in the computation order of the values in the program and is known as *forward*-mode AD. The opposite order of applying
the chain rule to the elemental operations, known as *reverse* or *adjoint*-mode AD, yields projections $\bar{\boldsymbol{x}} = \boldsymbol{J}^T \bar{\boldsymbol{y}}$.

This is achieved by applying to the elemental operations $\phi$, in reverse order of their original execution, the rule:

$$
\bar{a} = \bar{a} + \frac{\partial \phi}{\partial \bar{a}} \bar{a}; \quad \bar{b} = \bar{b} + \frac{\partial \phi}{\partial b} \bar{r}; \quad \ldots \bar{r} = 0 \quad .
\tag{3}
$$

where the bar operator defines the corresponding adjoint variable. For the multiplication example one therefore gets $\bar{a} = \bar{a} + b\bar{r}; \quad \bar{b} = \bar{b} + a\bar{r}; \quad \bar{r} = 0$. In particular, for applications in which $m \ll n$, the

---

[1] In practice most $\phi$ are uni- or bivariate.





reverse mode is advantageous because its computational cost does not depend on $n$. Typical problems in Cryosphere science involve computations of diagnostics which are scalar-valued cost functions ($m = 1$), such as for example the spatio-temporally averaged misfit between modeled surface

elevation and observed surface topography (Larour et al., 2014). For these cases, one can compute the gradient $\nabla f = \boldsymbol{J}^T \bar{y}$ with $\bar{y} = 1$ as a single projection. Thus, for high-resolution models implying very large $n$, the reverse mode is an enabling and potentially very efficient technique. This significant capability in AD is what makes its application to data assimilation so efficient. Instead of evaluating a Jacobian for each one of the outputs of the forward model with respect to each input, which would

be significantly consuming as it scales in n$^2$, the reverse mode evaluates the gradient of one specific output of interest with respect to the model inputs in only one sweep that scales in n.

Applying AD to large-scale frameworks such as ISSM (Larour et al., 2012), MITgm (Heimbach, 2008), SICOPOLIS (Heimbach and Bugnion, 2009) or DassFlow (Monnier, 2010) is a difficult proposition, but one which enables significant improvements in the way models can be initialized

(Heimbach and Bugnion, 2009), hindcast validated (Larour et al., 2012), and calibrated (Goldberg et al., 2015) towards better projections. Traditional approaches relying on Source-to-Source transformation have been developed, but for frameworks such as ISSM, which are C++ oriented, and highly parallelized, this type of approach breaks down. Our goal here is to demonstrate how the so-called operator-overloading AD approach can be implemented and validated for a framework such

as ISSM, and what developments were necessary to make this capablity operational. Our approach is discussed in section 2 of this manuscript, with section 3 describing the method validation as well as applications with ISSM. We discuss and conclude in the last section the applicability of such an approach to other frameworks, and on the opportunities these new developments afford for Cryosphere Science and data assimilation of remote sensing data in particular.

## 2 Methodology and Validation

### 2.1 Source to Source vs Overloaded Operators

Distinctions among the AD implementation approaches relate to the way the derivative data in $\dot{v}$ or $\bar{v}$ respectively is associated with the original program variable $v$ (data augmentation) and how the logic for coefficient propagation is added to the original program (logic augmentation). The basic options

for the former are *association by address* and *association by name*. Association by address packs the original and the derivative data into a new type called the *active* type, and all differentiated program variables have their type changed to that new active type. For the logic augmentation one uses *source code transformation* or *operator overloading*. Because of the complexity of the C++ syntax and semantics, there currently still is no comprehensive AD tool for source code transformation of C++

models and thus operator overloading remains the method of choice, implying association by address for the data augmentation. In practice, the overloaded operators and intrinsics for the forward mode





typically execute Eq. (2) directly for the forward mode and for the reverse mode record the sequence of operations into a trace $T(f(\boldsymbol{x}))$ which then is read backward by an interpreter $R(T)$ that executes the statements Eq. (3) for the reverse mode. The invocation of $R(T)$ is part of the user-written logic that then uses the derivatives. The model thus enhanced by an adjoint capability computing both $f$ and $\nabla f$ is denoted by $\overline{M}$.

### 2.2 Type Change to Enable Overloaded Operators

Changing to the aforementioned active type is a significant effort to be undertaken in the model code. Among the choices to effect this type change one should select one that is transparent to the model development process, is maintainable, and minimizes the manual effort.

The tool of choice for this paper is ADOL-C (Griewank et al., 1996; ADOL-C) and our target model $M$ is the Ice Sheet System Model (ISSM, Larour et al. (2012)). The ADOL-C manual is quiet on the practical approach to effect the type change. For models with a large C++ code base and many contributing developers such as ISSM, not all of them aware of $\overline{M}$ but merely interested in running $M$, it is important to make this process transparent and robust. The representation of $M$ as a computer program typically means that in the set of program variables $\mathcal{V} = \mathcal{A} \cup \mathcal{P}$ there is a subset $\mathcal{P}$ of *passive* variables that do not carry derivative information, such as variables of non-differentiable type (integers, strings) but also floating-point parameters, physical constants, or descriptors of the problem domain. The size of the trace $T$ is a major factor on the computational efficiency of the adjoint. Thus, one must categorize program variables into $\mathcal{A}$ and $\mathcal{P}$ with the aim to minimize the set $\mathcal{A}$ of active (i.e. type-changed) program variables. In ISSM this is accomplished by changing `double` variables to a type named for brevity e.g. `TA` (in ISSM `IssmDouble` ) for variables in $\mathcal{A}$ and to e.g. `TP` (in ISSM `IssmPDouble` ) for variables in $\mathcal{P}$, respectively. This categorization must be performed by an expert familiar with the global data dependencies in $M$ and the work that has been done for $\overline{M}$ and can be incrementally done for code contributions supplied as `double` variables without breaking $M$. This permits easy tracking as all `double` types should eventually be replaced with either `TA` or `TP` . Following good practices in C++ both `TA` and `TP` are `typedef` ed in a central location switched only there via a preprocessing macro to use the active type supplied by the AD tool or hide the distinction altogether for running the plain $M$. This implies that one cannot explicitly overload methods or define data structures for distinct `TA` and `TP` without filtering the declaration and definition of the `TA` variant by the preprocessor as well. Such code duplication introduces an undue code maintenance burden.

Instead, the recommended approach is the use of C++ template classes and methods where `TA` and `TP` are the concrete arguments for an abstract template type `T` . While the use of templates may imply a larger up-front effort if they weren't present in the model code before, as was the case in ISSM, the long term benefits not only for $\overline{M}$ but also the plain $M$ are obvious when one wants to experiment with computation at different precision levels. Aside from ISSM, an established general





example for this kind of templating is the use of SACADO within Trilinos (The Trilinos Project;
Phipps et al., 2008). Of particular value in ISSM was the migration of data containers for arrays

and (sparse) matrices and that of existing wrappers to `malloc` and `free` to templated wrappers
of `new` and `delete`, in ISSM called `xNew` and `xDelete`. The latter is necessary not only to
properly instantiate `TA` variables but also permits specializations for `TA` in $\overline{M}$ as explained in the
following.

Interpreting the trace by $R(T)$ means that the space holding the data for the variables $r, a, b \in \mathcal{V}^*$

(the set of instantiated program variables at run time) occurring in each operation $r = \phi(a, b)$ must
be represented by some mapping $\omega : \mathcal{V}^* \mapsto \mathbb{N}^+$ to pseudo addresses. In ADOL-C these addresses are
called *location*s and represent indices in a work array held by $R(T)$. The pseudo addresses must be
managed through the `TA` constructor and destructor in a fashion similar to the memory management
of the actual program variables themselves, i.e. pseudo addresses are assigned from and returned

to a pool $\Omega$ of available addresses. However, no distinction between heap and stack variables is
made and generally data locality will not be preserved. On the other hand, for special operations $\phi_A$
with array arguments of size $s$, that is, calls to external solvers (see section 2.3) or MPI routines (see
section 2.4), it would be counterproductive to record in $T$ the pseudo-addresses of each of the $s$ array
elements rather than a consecutive range. The latter, however, imposes that the pool be primed by

some call $\Omega(s)$ such that $\omega$ returns consecutive pseudo addresses when called by the constructor for
each element in the array. In ADOL-C this is done by calling `ensureContiguousLocations`
immediately before an active array is instantiated. Avoiding copying of array values in $M$ as an
efficiency measure means that any given array has a good chance of being used in $\phi_A$, and to avoid
littering the code with preprocessor-guarded calls to $\Omega(s)$, we decided in ISSM to instead add a call

to `ensureContiguousLocations` in the `TA` specialization of `xNew`:

```
#if defined(_HAVE_ADOL_C_)
template <>
adouble* xNew(unsigned int s) {
 ensureContiguousLocations(s);
 adouble* aT_p=new adouble[s];
 assert(aT_p);
 return aT_p;
}
#endif
```

Thus, the use of $\phi_A$ means frequent calls to $\Omega(s)$. The implementation of $\Omega(s)$ searches the pool
to find a sufficiently large consecutive range or otherwise grow the pool by at least $s$ addresses. In
ISSM, this search became a significant performance bottleneck. Minimizing the search effort and
balancing it with pool growth has been newly implemented in ADOL-C and is controlled by setting



parameters via `setStoreManagerControl` for the ratio of the addresses inside and outside of
the pool and the maximal pool size to trigger searches.

### 2.3 Binding to External Solvers

The strength of the AD approach lies in the automated differentiation of all the parts of the model
that are changing and expanding as the model is being developed as opposed to manually program-
ming adjoints which is error prone and implies a significant effort in terms of code development and
maintenance. However, because models reference libraries, for *fixed* mathematical mappings, that
may have easily programmable high-level adjoints, one can and should exploit this insight as it can
yield efficiency gains not obtainable by an AD tool remaining unaware that calls to some collection
of subroutines represent a certain mathematical mapping. A good example are linear solver libraries
that may not even be written in the same programming language as the model in question. Consid-
ering the system $As = r$, the solver $S$ computing $s = S(A, r)$ needs an adjoint counterpart solving
$A^T t = \bar{s}$, incrementing $\bar{r} = \bar{r} + t$ and, if $A \in \mathcal{A}$, also $\bar{A} = \bar{A} - \bar{b}s^T$. Clearly, the solve with the trans-
posed can easily be implemented reusing the solver $S$ as in $t = S(A^T, \bar{s})$. However, the answers to
the following questions affect the efficiency and ought to be considered in the context of the given
model:

Q1: Is $A \in \mathcal{A}$ which therefore requires the rank-1 update to $\bar{A}$?

Q2: Are the previous values of $s$, $r$ and $A$ overwritten by $S$ but used to compute partial derivatives
for Eq. (3) and therefore must be saved and restored?

Q3: If $S$ is a direct solver should one save the factors or refactorize for the adjoint solve?

Q4: How does the external function interface used by $R(T)$ allow for efficient reuse of intermediate
buffers?

External functions $f_e$ that had been supported by ADOL-C had the signature $f_e(l_x, x, l_y, y)$ with
inputs $x$, outputs $y$ for the original call, $l_x, l_y$ their respective array lengths, and $\overline{f_e}(l_y, \bar{y}, l_x, \bar{x})$ for
the adjoint counterpart. In the case of a linear system with $A \in \mathcal{A}$, the input $x$ is packed with both
$A$ and $r$ while $y$ contains the solution $s$ of the system on return. This, however, was insufficient for
binding to solvers from the GNU Scientific Library (GSL) (Galassi, 2009) and MUMPS (Amestoy
et al., 2001) used by ISSM. The following extensions were implemented in ADOL-C to enable the
ISSM adjoints but are generic in nature and would need to be supported in some form by other tools.

E1: Expanded the $\overline{f_e}$ interface by $x, y$ to $\overline{f_e}(l_y, \bar{y}, l_x, \bar{x}, x, y)$ to enable refactoring;

E2: added optional parameters to pass in the sparsity pattern of $A$ for MUMPS as a generic integer
array $i$ of length $l_i$ for both $f_e$ and $\overline{f_e}$;

E3: added controls for storing and restoring prior values of $x$ and $y$;



E4: added tracking of the maximal $l_x, l_y$ in sequences of $f_e$ calls.

Regarding Q1 we know that in ISSM $A \in \mathcal{A}$, and regarding Q2 the parameters passed to $f_e$ have no other uses and therefore, using the controls (E3), we avoid (re)storing their values. The direct

solver from the GSL used here had no API control to back solve with the factors for the transposed and we did not want to reverse engineer the permutation representation. Hence the refactoring was done as a matter of convenience for the sequential reference case requiring E1. The parallel and therefore practically more efficient MUMPS solver operates on sparse, distributed $A$, therefore requiring E2. MUMPS offers both the ability to store the factors to file and perform the back-solve for

the transpose. However, the MUMPS portion of the runtime is comparatively small (see section 3). Consequently the overhead for the file I/O when considering the factor data size after fill-in is not expected to yield much practical benefit in this context, answering Q3. Finally, for Q4, the extension E4 is exploited because transient runs of ISSM need to account for changes in the system size and a preallocation with the maximal buffer sizes therefore avoids some of the memory management

overhead.

### 2.4 Handling Parallelism with the AdjoinableMPI Library

As is the case with ISSM, practically relevant science problems incur a computational complexity that necessitates execution on parallel hardware, often using MPI as the paradigm of choice. Sending data with MPI from a source buffer $s$ to a target buffer $t$ can be interpreted as a simple assignment

$t = s$. This implies for the adjoint an increment $\bar{s} = \bar{s} + \bar{t}$, that is, the adjoint is a communication of the adjoint values in the reverse direction. The principles for adjoining two-sided MPI communication have been explored in Utke et al. (2009). The development of the AdjoinableMPI (AMPI) wrapper library AdjoinableMPI started in Fall of 2012. It is designed to provide an AD tool-independent implementation for adjoining MPI-parallelized C, C++ and Fortran models. The wrapper interfaces

distinguish themselves from the original MPI by the prefix `AMPI` and have a few additional parameters where needed to enable the adjoint functionality. AMPI also provides additional types and predefined symbols. Discussing the internal design of AMPI is outside the scope of this paper. However, the application to ISSM is the first large scale practical use of AMPI and in the following we will discuss the steps taken to use it in the ISSM code base.

For testing and small scale experiments, the ISSM code, as many other models, treats MPI parallelization as a compile-time option controlled by preprocessor macros. Furthermore, turning the adjoint capability on and off as suggested in section 2.2 would imply additional switching between the original MPI and AMPI with code duplication and the potential for errors. To avoid these undesirable consequences we decided to introduce in ISSM another wrapper layer (prefixed `ISSM_MPI`

) of calls and definitions to encapsulate completely the four functionality variants listed in Table 1.

The approach is shown for `MPI_Reduce` in Fig. 1. Variant 1 is implemented by line 21, variant 2 by lines 13-18 and 21, variant 3 by line 9, and variant 4 by line 7. The example code reflects some of



**Table 1.** Logic variants encapsulated in the ISSM MPI wrapper library

|   | MPI | AD |   |
|---|-----|----|----|
| 1 | no | no | emulate MPI semantic, e.g. with `memcpy` between buffers for `MPI_Reduce` |
| 2 | no | yes | emulate MPI semantic but for differentiation need to use the overloaded assignments instead of `memcpy` |
| 3 | yes | no | pass-through to plain MPI |
| 4 | yes | yes | call AMPI routines |

```
1  int ISSM_MPI_Reduce(void *sendbuf, void *recvbuf, int count,
2             ISSM_MPI_Datatype datatype, ISSM_MPI_Op op, int root,
3             ISSM_MPI_Comm comm){
4        int rc=0;
5  #ifdef _HAVE_MPI_
6  #ifdef _HAVE_AMPI_
7        rc=AMPI_Reduce(sendbuf, recvbuf, count, datatype, op, root, comm);
8  #else
9        rc=MPI_Reduce(sendbuf, recvbuf, count, datatype, op, root, comm);
10 #endif
11 #else
12 #ifdef _HAVE_ADOLC_
13        if (datatype==ISSM_MPI_DOUBLE) {
14          IssmDouble* activeSendBuf=(IssmDouble*)sendbuf;
15          IssmDouble* activeRecvBuf=(IssmDouble*)recvbuf;
16          for(int i=0;i<count;++i) activeRecvBuf[i]=activeSendBuf[i];
18        }
19        else
20 #endif
21            memcpy(recvbuf,sendbuf,sizeHelper(datatype)*count);
22 #endif
23     return rc;
24 }
```

**Figure 1.** Code for wrapping reduction

the simplifying assumptions being made based on the MPI usage patterns in ISSM. Examples are the absence of `MPI_IN_PLACE` and user-defined MPI types as well as all active data being of double precision. This permits the simple distinction on line 12 but of course the logic covers all other non-differentiated MPI types that may occur. The following lines 14,15 indicate the pairing between the C++ type definition `IssmDouble` and the wrapper-defined `ISSM_MPI_DOUBLE` to signify active data being communicated when AD is enabled. Mirroring in MPI the approach described in section 2.2 with two types `TA` and `TP`, the passive type `TP` (concrete `IssmPDouble`) has an MPI





**Table 2.** Types per variant from Table 1; * active type from ADOL-C, ** active MPI type from AMPI

|   | ISSM_MPI_DOUBLE<br>#define to | IssmDouble (=TA)<br>typedef to | ISSM_MPI_PDOUBLE<br>#define to | IssmPDouble (=TP)<br>typedef to |
|---|---|---|---|---|
| 1 | 2 | double | 3 | double |
| 2 | 2 | adouble * | 3 | double |
| 3 | MPI_DOUBLE | double | MPI_DOUBLE | double |
| 4 | AMPI_ADOUBLE ** | adouble * | MPI_DOUBLE | double |

counterpart defined in the wrapper as `ISSM_MPI_PDOUBLE`. It too can be the value passed in via
the `datatype` argument at line 1 and signals to AMPI passive floating point communications. The
definitions provided by the `ISSM_MPI` wrapper corresponding to types `TA` and `TP` are given in
Table 2.

The matching between the actual type of the buffer and the corresponding MPI type must extend
to the templating approach suggested for the type change in section 2.2. The MPI standard keeps the
definition of `MPI_Datatype` opaque and MPI types can be created at runtime, that is, the value
may not be a compile time constant usable as a value template parameter. Therefore, a template
buffer type `T` must be paired with the corresponding MPI type value declared as a pointer parameter.
The approach is shown in Fig. 2. Assuming many AMPI calls in `foo`, this reduces the count of code
locations where type errors may be introduced to the template instantiations.

```
// this declaration comes from the wrapper:
extern ISSM_MPI_Datatype ourISSM_MPI_DOUBLEVal, ourISSM_MPI_DOUBLEVal;

template <class T, ISSM_MPI_Datatype *mt_p> void foo(T *t){
  ISSM_MPI_Reduce(t, ..., ,*mt_p,..);
}

void bar(IssmDouble *x, IssmPDouble *p) {
  foo<IssmDouble,&ourISSM_MPI_DOUBLEVal>(x);
  foo<IssmPDouble,&ourISSM_MPI_PDOUBLEVal>(p);
}
```

**Figure 2.** Code snippets for templating with corresponding MPI data types

Finally, a practical concern for using the parallelized adjoint is the handling of sensitivities to
quantities that are uniformly initialized across ranks, such as parameters. Frequently, as was the
case within ISSM, these quantities are initialized from files or otherwise per process in the parallel
case the same way as in the sequential case. In the parallel case that implies a replication of the
same quantity across ranks. However, to obtain the correct sensitivities, the quantity $q$ in question,





should be unique, in other words that quantity must be uniquely initialized at one root rank and then broadcast to the other ranks. Otherwise, for $r$ ranks, then at each rank one would obtain only a part $\bar{q}_i$ of the total $\bar{q}$ and would have to "manually" sum up $\bar{q} = \sum \bar{q}_i$. With an initial broadcast of $q$, however, the corresponding adjoint provided through AMPI by using `AMPI_Bcast` is that exact

sum reduction and $R(T)$ yields the correct adjoint at the broadcast root. This notion similarly applies to any situation where a conceptually unique quantity of active type is implicitly replicated on some ranks.

### 2.5   Validation

ISSM is validated in AD mode by continuously running a test suite within the Jenkins (Smart, 2011)

integration and development framework (available here at http://issm.jpl.nasa.gov/developers/) . A detailed description of the suite of benchmarks is given in Table 3) The aim is to 1) compare forward runs in ISSM with their counterparts when overloaded operators are switched on. The results should be identical within double precision tolerances; and 2) compare forward and reverse runs carried out with ISSM AD on and off, using the GSL and MUMPS solvers. Comparisons of gradients

computed in AD mode with standard forward differences methods are also carried out to make sure the gradients computed (essentially in reverse scalar mode, which is the mode of predilection in ISSM for data assimilation) in AD mode are accurate.

## 3   Application

### 3.1   Application

The ongoing use of adjoint computations includes sensitivity studies and state estimation problems for transient model runs. Because this paper concentrates on technical aspects, we show, only as exemplary evidence of the practical usefulness, some sensitivities of cost functions calibrated for two sensors commonly encountered in Cryosphere Science, altimeters (that measure surface elevation), and radars (that measure surface displacement, or velocity). In Fig. 3, the sensitivity (gradient) of a

cost function related to ICESat-1 altimetry with respect to Surface Mass Balance (SMB) is shown for three epochs, September 2003, June 2006 and February 2009. The cost function is the spatio-temporal average of the misfit between modeled surface elevations (from the mass-transport module of the transient solution in ISSM), and the corresponding ICESat-1 altimetry record. This cost function was used within an inversion method to reconstruct SMB over the entire ICESat-1 time record

(Larour et al., 2014).

A second type of observations garnering considerable interest is temporally resolved time series of radar observations (using speckle-tracking, or InSAR to infer surface deformation of the ice) to measure variations of surface velocity on a seasonal time scale (one observation every 2 months, as in Moon et al. (2015)). Temporally inverting for such time series, trying to reconstruct for example



**Table 3.** ISSM AD validation suite integrated within Jenkins for continuous integration and delivery (Smart, 2011). Tests 3001 to 3010 and 3101 to 3110 test the repeatability of forward runs with and without ADOL-C compiled, but with no AD drivers specifically called. The forward runs involved are the standard stress balance, mass transport and thermal solutions, with 2D SSA (Shelfy-Stream Approximation MacAyeal (1989)), 3D SSA, 3D HO (Higher-Order, (Blatter, 1995; Pattyn, 1996)), 3D Full-Stokes (Stokes, 1845) and DG (Discontinuous-Galerkin) formulations. Test 3015 tests the AD GSL capability, by comparing the AD Forward Scalar mode (where we compute the gradient of ice volume with respect to ice thickness at one vertex of the mesh) against a simple forward differences computation on the same gradient. Test 3019 validates the AD GSL capability in Reverse Scalar mode (where we compute ice volume gradient with respect to thickness at all vertices of the mesh) vs the Forward Vectorial mode. Both gradients should be identical. Finally, test3119 validates the parallel capabilities (AD MUMPS) by comparing reverse scalar computations with GSL and MUMPS.

| Test | Solution Sequence | Formulation | Solver | Description |
|------|-------------------|-------------|--------|-------------|
| 3001/3101 | Stress Balance | 2D SSA | GSL/MUMPS | Equal runs with overload on and off |
| 3002/3102 | Stress Balance | 3D SSA | GSL/MUMPS | Equal runs with overload on and off |
| 3003/3103 | Stress Balance | 3D HO | GSL/MUMPS | Equal runs with overload on and off |
| 3004/3104 | Stress Balance | 3D FS | GSL/MUMPS | Equal runs with overload on and off |
| 3005/3105 | Mass Transport | 2D | GSL/MUMPS | Equal runs with overload on and off |
| 3006/3106 | Mass Transport | 2D (DG) | GSL/MUMPS | Equal runs with overload on and off |
| 3007/3107 | Mass Transport | 3D | GSL/MUMPS | Equal runs with overload on and off |
| 3008/3108 | Thermal Steady State | 3D | GSL/MUMPS | Equal runs with overload on and off |
| 3009/3109 | Thermal Transient | 3D | GSL/MUMPS | Equal runs with overload on and off |
| 3010/3110 | Transient 2D | 2D | GSL/MUMPS | Equal runs with overload on and off |
| 3015 | Mass Transport 2D | 2D | GSL | AD Forward Scalar vs Forward Differences |
| 3019 | Thermal Transient | 3D | GSL | AD Reverse Scalar vs AD Forward Vectorial |
| 3119 | Thermal Transient | 3D | MUMPS vs GSL | AD Reverse Scalar (MUMPS vs GSL)Scalar |

basal friction is a topic of interest due to its relevance in terms of understanding the dynamics of calving, basal slip and shear softening, and associated feedback mechanisms. In Fig. 4, we demonstrate the feasibility of such inversions by computing the gradient of a cost function related to the Moon et al. (2015) time series with respect to basal drag at epochs 2008, 2010 and 2013. The cost function is the spatio-temporal average of the misfit between modeled surface velocity (from the stress-balance module of the transient solution in ISSM) and observed surface velocities.

### 3.2 Benchmarking

Currently, these adjoint computations are performed on the Pleiades cluster at the NASA Advanced Supercomputing Center. Given the theoretical invariance of the effort for computing the adjoint $\nabla f$ with respect to the gradient size $n$ as stated in section 2, the practical question remains about the actual runtime overhead of computing both $f$ and $\nabla f$ when compared to computing just the original





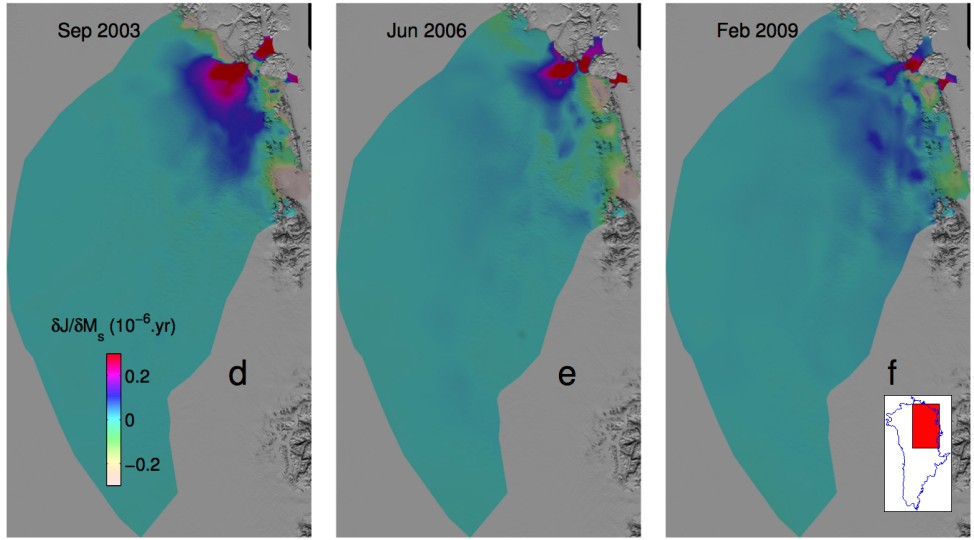

**Figure 3.** Gradient of surface altimetry cost function (spatio temporal average of the misfit between the 6 year record of observed ICESat altimetry and modeled surface elevation) with respect to SMB (in years). The gradient is computed for every time step of a two-week interval time series between 2003 and 2009. We only show three periods (September 2003, June 2006 and February 2009) in the entire interval. The location for the study is North-East Greenland Ice Stream, and the gradient is taken from an inversion study of the surface forcings necessary to best fit the ICESat altimetry record (Larour et al., 2014).

$f$. Here in particular one considers the effects of compiler optimization on the original code for $f$ using the built-in floating point operations on one side and in contrast to that the overhead incurred by calling the overloaded operators as well as the creation, storage, and interpretation of the trace $T$ (as the means to compute $\nabla f$) on the other. Aside from the fact that the tuning of the ISSM adjoint is

a work in progress we want to highlight the overwhelming impact of the application specific aspects on the runtime ratio. Therefore, we emphasize that exhibiting any particular overhead number as the ultimate result of scenario-specific tuning is of little use to the practitioner wanting to answer science questions. Rather, using examples from the ISSM work we want to show what may prevent achieving a satisfactory overhead.

The earliest ISSM adjoint computations took place before the MPI wrapper library was introduced and therefore were done sequentially with the GSL solvers. To evaluate the performance, a representative test case was chosen from the ISSM regression test suite (test 101, modeling the stress-balance of a square ice shelf). The problem size was indirectly set by specifying a measure for the maximal resolution of the generated mesh, thus increasing the number of mesh elements generated for a

smaller resolution parameter. The overhead factors are shown separately for the overloading as such and the generation of $T$ and $R(T)$ as wall-clock comparisons to the unmodified model compiled



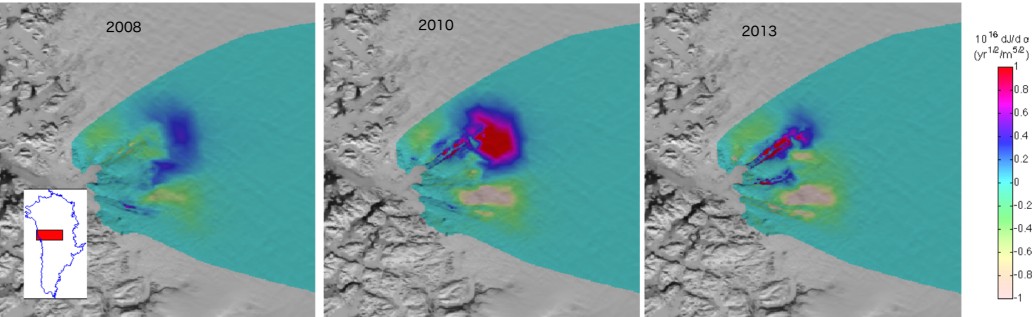

**Figure 4.** Gradient of surface velocity cost function (spatio temporal average of the misfit between a 5 year time series of observed surface velocities and the modelled surface velocity) with respect to the basal drag coefficient ( in $yr^{1/2}/m^{5/2}$). The gradient is computed for every time step of a two-week interval time series between 2008 and 2013. We only show three periods (2008, 2010 and 2013) in the entire interval. The location for the study is Upernavik Glacier, Central West Greenland, and the gradient is taken from an inversion study of the basal forcings required to match observed RADAR time series of surface velocities and calving front positions (*Larour et al, 2016, in preparation*).

with default optimization (-O2) in Fig. 5 (upper frame). While this plot indicates a small overhead factor of $\approx 4.5$ in particular for the largest mesh case (distance 12.5 km) the reason for this becomes apparent in the plot on the lower frame. It shows that the majority of the run time is consumed by

the GSL solver (libgsl) completely overshadowing any of the overhead caused by the adjoint for the largest mesh. We want to emphasize that GSL was chosen not for its efficiency but for the simplicity of the setup which quickly enabled adjoint computations. Introducing AMPI and thereby moving to a more appropriate solver (MUMPS) causes the adjoint overhead to become more prominent. The most consequential change necessitated by the use of MPI is the forced contiguity of the pseudo

addresses (see section 2.2).



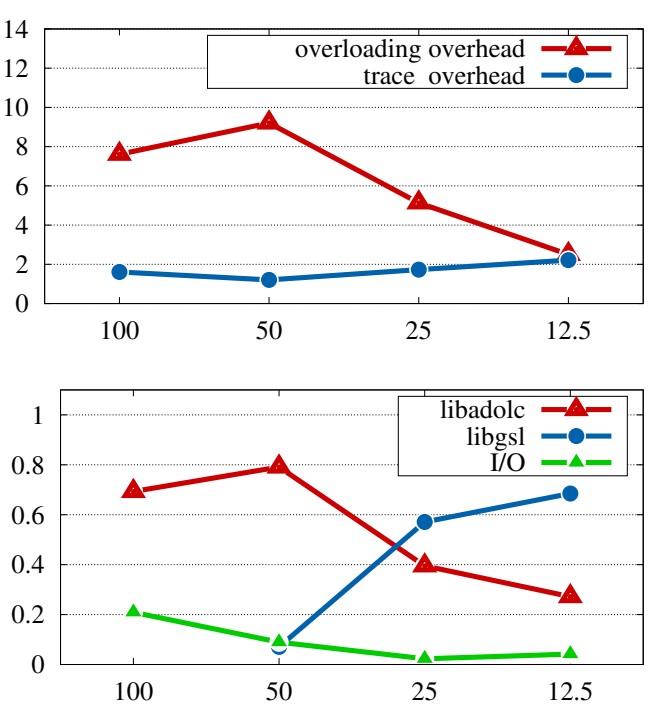

**Figure 5.** Test problem setup with sequential GSL solver; plots over max mesh distance show overhead factors (upper frame), and approximate run-time portions of the model execution with adjoint computation (lower frame)




In combination with the much reduced impact of the solver, the overhead factor for equivalent test problems reached temporarily up to 145 which clearly was not acceptable. Subsequent changes to the ADOL-C tool included improvements in the internal address management. Changes to the model included modifications of the sparse data format, enabling the control of the I/O buffering for

the trace $T$ and the ADOL-C address manager via the ISSM configuration specific to the setup to be computed. The combination of these changes led to regaining better performance and effective overall overheads between 10 and 30. Analyzing the details of the performance shows that the overhead factor for the trace creation and interpretation does not change significantly (see Fig. 6 bottom) with the mesh size in accordance with the theoretical result for the adjoint computation. The large major-

ity of the total overhead, evidenced by the runtime portion for libadolc in Fig. 6 (top) still originates in the internal address management. While the overall overhead factors are sufficiently small for the practical use of the adjoints for science problems further improvements in the addressing scheme are clearly warranted and subject of ongoing work.

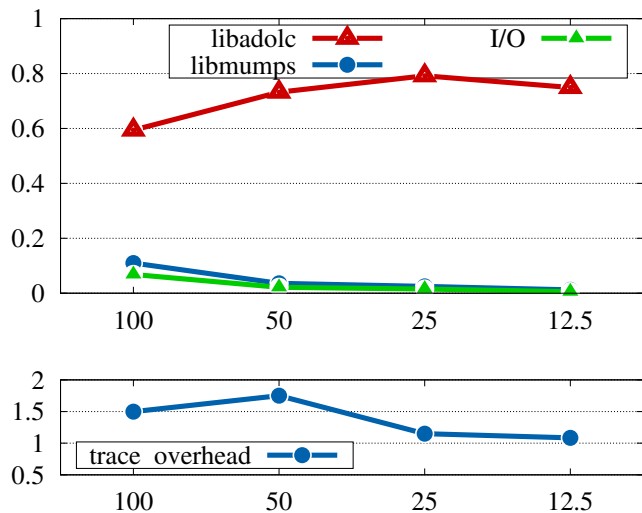

**Figure 6.** Test problem setup with parallel MUMPS solver; plots over max mesh distance show approximate run-time portions of the model execution with adjoint computation (upper frame) and overhead factors for the trace creation and interpretation (lower frame)



## 4   Conclusions

We developed a new adjoint capability in ISSM, based on the ADOL-C framework and the Adjoin-ableMPI library, which is to our knowledge the first time this type of approach has been systemat-ically applied to a software framework of this size and complexity. Despite the difficulties encoun-tered rewriting the software, the overloaded approach is transparent to the user, which is critial given the size of the larger Cryosphere Science commuity that is not familiar with the adjoint work, and for

which classic approaches such as source-to-source transformation have proven to be overly cumber-some. The flexibility of this approach allows in particular for quick turn-around in developing adjoint models of new parameterizations which are not easily hand derived. This is a major advantage in that it opens this approach to the wider community. This, given the large amount of remote sensing data currently being collected and under-utilized, could prove paramount if we are to hindcast validate

projections of sea-level rise. Further work is of course required to bring in additional observations such as gravity sensors, or radar stratigraphy observations, which will involve development of new cost functions, and scalability in 3D. Though this is complex in that it requires integrated resiliency and adjoint checkpointing schemes for long running transient modeling scenarios, our approach has proven flexible, and should lead to a brand new set of data assimilation capabilities that have already

been available to other Earth Science communities for a long time. Indeed, by allowing temporal data assimilation for a large number of sensors and models, such as demonstrated here with the use of altimetry and radar sensors for mass transport and stress balance models respectively, ISSM paves the way for wider integration between the modeling and observational Cryosphere community.

## 5   Code Availability

The ISSM code and its AD components are available at http://issm.jpl.nasa.gov. The instructions for the compilation of ISSM in AD mode, along with test cases is presented in the supplement attached to this manuscript.

*Acknowledgements.* DE-AC02-06CH11357. Larour and Utke were supported by the Jet Propulsion Labora-tory, California Institute of Technology under a contract with the NASA Cryospheric Sciences and Modeling and Analysis Program. Gilberto Perez was supported by a subcontract from the Jet Propulsion Laboratory to University of California at Irvine and Mathieu Morlighem was supported under a contract with the NASA IceBridge Research Program.





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
