# Peer review of "An Approach to Computing Discrete Adjoints for MPI-Parallelized Models Applied to the Ice Sheet System Model 4.11"

_Geoscientific Model Development, 2016_

## Referee Comment (RC1) · L. Hascoet (Referee) · 27 May 2016

General comments:

The paper describes the construction of the adjoint of the ice sheet simulator ISSM (written in C++) through the use of the overloading-based AD tool ADOL-C, and through the use of the MPI wrapper library AMPI (Adjoinable MPI). The paper presents the first meaningful gradients obtained with this adjoint ISSM, and discusses performance aspects. The main focus of the paper is about the programming strategy that was chosen to ease the algorithmic differentiation process with ADOL-C, the specific efficient tactic to propagate the adjoint through calls to library linear solvers, and the constraints and benefits of using th AMPI library to get an MPI-parallel adjoint.

Although not a specialist of the application domain, I have the feeling that the main similar efforts by glaciology researchers are presented and compared adequately. The paper claims that the adjoint ISSM will allow for full exploitation of the enormous quantity of measurement data available. This is something one can believe in.

The strategies proposed in sections 2.2 to 2.4 go into details that shed an interesting light on ADOL-C (questions of location handling and contiguity that I was not aware of, thanks) and on the technicalities of the linear solver "trick". Many of these are of general interest for the AD people, overloaders or source-transformers alike.

Text is well written, in excellent English. My only minor concern is about its high level of technicity in Computer Science, which might make it hard to read at places, especially for readers specialized rather in geophysics models. But this high level makes it a profitable read for computer science (in particular AD) people.

To summarize, I like the paper. I think it is of value for developers of models, and for AD tool developers as well. Getting this value requires effort while reading, though.

Specific comment:

Still, we face this old dilemma between source-transformation AD (probably more efficient) applied to codes in old-fashioned languages (Fortran or C), versus overloading-based AD (slightly heavier in memory) applied to codes in modern languages (nicer to develop). Sentence at line 370 obviously gives the authors' opinion on that, but I'm wondering if things are really that terrible. After all, in other application domains, people do use source-transformation AD (when available).

On APMI in this context, the paper might give more hints on what happens while interpreting the trace R(T) i.e. when an AMPI call is registered in T, R(T) calls AMPI primitives at that time too. Otherwise, the reader might wonder how this can work.

Technical corrections:

– Line 19 "alreay"

**[GMDD](about:blank)**
– I don't understand the nˆ2 vs n at line 90. Am I missing something?

– Sentence at line 140 look ambiguous to me. Maybe a few commas or small words would help?

– Using * and ** in table 2 is unfortunate: guaranteed confusion with the C dereference operator.

– Paragraph at lines 269-275 is hard to follow. Can it be rephrased ?

– Line 368 "critial"

––––––––––––––––––––––––––––––

---

## Short Comment (SC1) · 6 Jun 2016

Dear authors,

In my role as Executive editor of GMD, I would like to bring to your attention our Editorial version 1.1:

http://www.geosci-model-dev.net/8/3487/2015/gmd-8-3487-2015.html

This highlights some requirements of papers published in GMD, which is also available on the GMD website in the 'Manuscript Types' section:

http://www.geoscientific-model-development.net/submission/manuscript_types.html

In particular, please note that for your paper, the following requirement has not been met in the Discussions paper:

- "The main paper must give the model name and version number (or other unique identifier) in the title."

Please add a version number for the ISSM model in the title upon your revised submission to GMD.

Yours,

Astrid Kerkweg

---

## Referee Comment (RC2) · Anonymous Referee #2 · 15 Jun 2016

The paper illustrates how Algorithmic Differentiation (AD) is implemented in the Ice Sheet System Model (ISSM) to compute discrete adjoints and sensitivities. The paper details the operator overloading approach used to achieve AD without disrupting the existing forward capability. Further it explains the parallelization is performed. Overhead factors for computing adjoints are reported for a couple of model runs.

Although the topic of the paper is surely interesting for this journal, I think the exposition is not always clear. In particular, in the introduction there are some typos, inaccuracies and undefined quantities. I would recommend the authors to revise the paper, particularly the introduction, in order to make it more understandable to the reader not used to forward and reverse AD. The paper focuses on several (important) technical details

but I think it lacks a mathematical/algorithmic description of the overall framework.

I would recommend the authors to consider a typical sensitivity problem encountered in applications and show how it is solved in ISSM by reverse AD. An exemplar problem could be to find the sensitivity (derivative) of a functional $f = f(x, p)$ with respect to the parameter $p$, subject to the (nonlinear) constraint $c(x, p) = 0$ (here $f$ could represent a misfit of some quantity with observational data, and $c$ the ice sheet model). How this problem (or a similar one, maybe time dependent) is solved by ISSM using revers AD? What linear/nonlinear systems need to be solved?

Some ice-sheet solvers compute sensitivities by solving an additional adjoint equation: BISICLES (Cornford at al. JCP, 2013), Ymir(Isaac et al, SISC, 2015), Albany/FELIX (perego et.al, JGR, 2014 and Tezaur et al. GMD, 2015). Albany/FELIX computes the partial derivatives of the constraint and functional w.r.t. the parameters and the independent variables using forward AD (Pawlowski et al, Scientific Programming, 2012). A different AD approach is used in Goldberg et al., GMD, 2016. How does the ISSM approach compares to these?

**0.0.1 Detailed Review**

Line 45 How can automatic differentiation be possible when a manual differentiation is not possible? AD is a way to perform differentiation, so if a model is not differentiable then AD would give NaN or Inf.. or similar.

Line 63 If the expression to be differentiated is implicit than the accuracy of the solution would depend on the implicit solvers and would mostly likely be larger than machine precision.

Line 73 Variable $\mathbf{y}$ has not been defined. I assume is $\mathbf{y} = \mathbf{f}(\mathbf{x})$.

Eq. 3 I think there is a typo here. Should it be

$$\bar{a} = \bar{a} + \frac{\partial \phi}{\partial a}\bar{r}; \quad \bar{b} = \bar{b} + \frac{\partial \phi}{\partial b}\bar{r} \quad ?$$

And why $r = 0$? Please check also expressions at line 81. I think this part should be expanded and the reverse AD method should be presented more clearly.

Line 82 The number of sweeps depends only on $m$ but the computational cost depends on $n$ as well (as mentioned at line 91).

Line 86 In this instance what does $f$ denote? The misfit in the surface elevation? And $x$? Please describe better in the paper the parallel between the physical problem and the equation.

Line 201 How's $\bar{b}$ is defined? Please expand a bit this part to help the reader better understand how an implicit equation as $As = r$ is handled by reverse AD. Also, in the introduction $r$ was used to denote an elemental operation (or its output), whether here it is used to denote the residual of the system, which is an input of the "elemental" operation $s = A^{-1}r$, I think this can be confusing. Let's say only a few elements of $A$ are active (say, the part related to the boundary, if A comes from discretizing a PDE), then do you need to store the entire matrix $\bar{A}$ as well? What happens when the operator $A$ is non linear (it depends on $s$)?

Line 324 Again, I don't think that the computational cost do not depend on $n$, even theoretically.

Figure 5-6 What is the specific application solved to produce results showed in Figures 4 and 5? Are these nonlinear problems? How many processors are you using when solving the probem with MUMPs? Please add also the overloading overhead in Figure 6, as done in Figure 5.

---

## Author Comment (AC1) · 15 Sep 2016

**1 Editor Comment #1**

Dear authors,
In my role as Executive editor of GMD, I would like to bring to your attention our Editorial version 1.1:
http://www.geosci-model-dev.net/8/3487/2015/gmd-8-3487-2015.html
This highlights some requirements of papers published in GMD, which is also available on the GMD website in the Manuscript Types section:
http://www.geoscientific-model-development.net/submission/manuscript_types.html
In particular, please note that for your paper, the following requirement has not been met in the Discussions paper:

- "The main paper must give the model name and version number (or other unique identifier) in the title."

Please add a version number for the ISSM model in the title upon your revised submis- sion to GMD.

Yours,

Astrid Kerkweg

We thank the executive editor for catching this issue, and have accordingly added the ISSM version number to the title.

**2 Reviewer #1: L. Hascoet**

General comments:

The paper describes the construction of the adjoint of the ice sheet simulator ISSM (written in C++) through the use of the overloading-based AD tool ADOL-C, and through the use of the MPI wrapper library AMPI (Adjoinable MPI). The paper presents the first meaningful gradients obtained with this adjoint ISSM, and discusses performance aspects. The main focus of the paper is about the programming strategy that was chosen to ease the algorithmic differentiation process with ADOL-C, the specific efficient tactique to propagate the adjoint through calls to library linear solvers, and the constraints and benefits of using th AMPI library to get an MPI-parallel adjoint. Although not a specialist of the application domain, I have the feeling that the main similar efforts by glaciology researchers are presented and compared adequately. The paper claims that the adjoint ISSM will allow for full exploitation of the enormous quantity of measurement data available. This is something one can believe in. The strategies proposed in sections 2.2 to 2.4 go into details that shed an interesting light on ADOL-C (questions of location handling and contiguity that I was not aware of, thanks) and on the technicalities of the linear solver

"trick". Many of these are of general interest for the AD people, overloaders or source-transformers alike. Text is well written, in excellent English. My only minor concern is about its high level of technicity in Computer Science, which might make it hard to read at places, especially for readers specialized rather in geophysics models. But this high level makes it a profitable read for computer science (in particular AD) people. To summarize, I like the paper. I think it is of value for developers of models, and for AD tool developers as well. Getting this value requires effort while reading, though.

We thank Dr. Hascoet for the positive assessment of the manuscript, in particular concerning the applicability of the approach described to the wider Cryosphere community, as well as the AD community. As such, this manuscript is indeed faced with the difficulty of being quite technical. We have therefore strived in this round of edits to clarify, if not simplify, the manuscript where it was possible to do so, without taking away the necessary technicality required for application to the AD community in particular.

Specific comments:

Still, we face this old dilemma between source-transformation AD (probably more efficient) applied to codes in old-fashioned languages (Fortran or C), versus overloading- based AD (slightly heavier in memory) applied to codes in modern languages (nicer to develop). Sentence at line 370 obviously gives the authors opinion on that, but Im wondering if things are really that terrible. After all, in other application domains, people do use source-transformation AD (when available).

The point is well taken, as indeed source-transformation AD in other application domains has gained acceptance. In Cryosphere though, this is far from being the case one could argue. We understand the subjectivity of our point of view on this matter, and have therefore rephrased this part of the conclusion accordingly.

On APMI in this context, the paper might give more hints on what happens while interpreting the trace R(T) i.e. when an AMPI call is registered in T, R(T) calls AMPI primitives at that time too. Otherwise, the reader might wonder how this can work.

Thank you for pointing this issue. We have addressed it by modifying line 240 in the manuscript from "This implies for the adjoint an increment s= s+t, that is, the adjoint is a communication of the adjoint values in the reverse direction." to "This implies for the adjoint an increment s= s+t, that is, the adjoint of a communication is a reversal of the data flow between buffers. Calculating the adjoint of t=s means sending back the value of t and computing s= s+t."

Technical corrections:

- Line 19 "alreay": typo corrected.

- I dont understand the n2 vs n at line 90. Am I missing something? The manuscript from line 80 to line 91 is indeed misleading, and we have addressed this by removing the concept of scaling. The issue is rather how many forward or backward sweeps are necessary to compute the Jacobian components of interest. We have rephrased the paragraph accordingly. This also proved to be confusing to reviewer #2, and should therefore also resolve his concerns.

- Sentence at line 140 look ambiguous to me. Maybe a few commas or small words would help? The sentence is indeed long and convoluted, we broke it down into two simpler sentences.

- Using * and ** in table 2 is unfortunate: guaranteed confusion with the C dereference operator. Agreed, we have replaced * by +.

- Paragraph at lines 269-275 is hard to follow. Can it be rephrased ? We rephrased the paragraph to try and simplify it, in particular, why the MPI_DataType cannot be used as a reliable value for template parameterization, as it's not well defined at compile time.

- Line 368 "critial". typo corrected.

**3   Reviewer #2**

The paper illustrates how Algorithmic Differentiation (AD) is implemented in the Ice Sheet System Model (ISSM) to compute discrete adjoints and sensitivities. The paper details the operator overloading approach used to achieve AD without disrupting the existing forward capability. Further it explains the parallelization is performed. Overhead factors for computing adjoints are reported for a couple of model runs.

Although the topic of the paper is surely interesting for this journal, I think the exposition is not always clear. In particular, in the introduction there are some typos, inaccuracies and undefined quantities. I would recommend the authors to revise the paper, particularly the introduction, in order to make it more understandable to the reader not used to forward and reverse AD. The paper focuses on several (important) technical details but I think it lacks a mathematical/algorithmic description of the overall framework.

I would recommend the authors to consider a typical sensitivity problem encountered in applications and show how it is solved in ISSM by reverse AD. An exemplar problem could be to find the sensitivity (derivative) of a functional f = f (x, p) with respect to the parameter p, subject to the (nonlinear) constraint

c(x, p) = 0 (here f could represent a misfit of some quantity with observational data, and c the ice sheet model). How this problem (or a similar one, maybe time dependent) is solved by ISSM using revers AD? What linear/nonlinear systems need to be solved?

We thank the reviewer for the time spent on the review, and for the legitimate points raised in the manuscript. In particular, the focus on the clarity of the manuscript in the introduction, and the lack of a mathematical framework. The idea of considering a typical sensitivity problem is a propos, as it is exactly what we have shown in Figs 3 and 4. However, we have clearly not conveyed this efficiently. We have therefore reworked the introduction to better explain the AD approach to the non-cognizant scientist, improved or clarified the explanation of the mathematical framework in Eqs. 1 to 3, and more explicitely linked the framework to the examples exposed in Figs. 3 and 4 for North-East Greenland Ice Stream and Upernavik Glacier, which fit exactly the type of non-linear, transient, sensitivity analyses called for by the reviewer as examples.

Some ice-sheet solvers compute sensitivities by solving an additional adjoint equation: BISICLES (Cornford at al. JCP, 2013), Ymir(Isaac et al, SISC, 2015), Albany/FELIX (perego et.al, JGR, 2014 and Tezaur et al. GMD, 2015). Albany/FELIX computes the partial derivatives of the constraint and functional w.r.t. the parameters and the inde- pendent variables using forward AD (Pawlowski et al, Scientific Programming, 2012). A different AD approach is used in Goldberg et al., GMD, 2016. How does the ISSM approach compares to these?

We thank the reviewer for pointing towards this body of litterature. We have included it in the introduction, and compared against our ISSM approach when necessary.

Detailed Review:

- Line 45: How can automatic differentiation be possible when a manual differentiation is not possible? AD is a way to perform differentiation, so if a model is not differentiable then AD would give NaN or Inf.. or similar. The point is well taken. The issue relates more to the feasibility of manual differentiation, which can become exponentially prohibitive. We have rephrased accordingly.

- Line 63: If the expression to be differentiated is implicit than the accuracy of the solution would depend on the implicit solvers and would mostly likely be larger than machine precision. We agree with this point, and have rephrased this sentence to convey the fact that the precision reached by the AD approach is on par with the precision dictated by the implicit solvers themselves.

- Line 73: Variable y has not been defined. I assume is y = f(x). Thanks for catching this, we have corrected the issue in Eq. 1.

- Eq. 3: I think there is a typo here. Should it be a = a + r ; b = b + r ? a b There is definitely a typo, thanks for catching it.

- And why r = 0? Please check also expressions at line 81. I think this part should be expanded and the reverse AD method should be presented more clearly. This part has been redesigned, as it was not clear indeed. Here is the new paragraph in the manuscript:
  "This is achieved using the partial derivatives of the forward operation $\phi$ as weights in redistributing the adjoints of the result $r$ to the adjoints of the respective arguments $a$, $b$ following the rule

$$\bar{a} = \bar{a} + \frac{\partial \phi}{\partial a}\bar{r}; \;\; \bar{b} = \bar{b} + \frac{\partial \phi}{\partial b}\bar{r}; \;\; \ldots \bar{r} = 0 \quad . \tag{1}$$

  Here $\bar{v}$ denotes the adjoint corresponding to variable $v$. The final zeroing of the result adjoint $\bar{r}$ corresponds to the notion that $r$ had been assigned a new value overriding any previous value $r$ might have had. Correspondingly the adjoint of $r$ has been completely distributed and to allow the increment formulation for the adjoint of any prior value of $r$ it has to (re)set to 0."

- Line 82 The number of sweeps depends only on m but the computational cost depends on n as well (as mentioned at line 91). We totally agree. We refer to the same concern raised by Dr. Hascoet, which we already addressed.

- Line 86: In this instance what does f denote? The misfit in the surface elevation? And x? Please describe better in the paper the parallel between the physical problem and the equation. Here, f refers to the function described in Eq. 1. We have corrected the manuscript accordingly. In terms of overall manuscript rewrite, we have now better defined the applications shown in Figs. 3 and 4 (the physical problem) and the description of the problem from Eqs. 1 through 3, as requested by the same reviewer in the general remarks also.

- Line 201: Hows b is defined? Please expand a bit this part to help the reader better understand how an implicit equation as As = r is handled by reverse AD. Also, in the introduction r was used to denote an elemental operation (or its output), whether here it is used to denote the residual of the system, which is an input of the el- emental" operation s = A1r, I think this can be confusing. Lets say only a few elements of A are active (say, the part related to the boundary, if A comes from discretizing a PDE), then do you need to store the entire matrix A as well? What happens when the operator A is non linear (it depends on s)? There is a typo, and b is now replaced by r. Indeed the choice of r vs b is unfortunate, and

we have reverted to b by default here. In terms of operation, the As = r operation represents an elementary operation, and does not represent a non-linear system of equations per se. This also mitigates the question from the reviewer regarding boundary conditions: the matrix A here is the matrix corresponding to the free degrees of freedom, after reduction from the global set of degrees of freedom by constraining the degrees of freedom lying on the Dirichlet boundary conditions of the system. But it could equally well correspond to any equation system As=r where A is not a stiffness matrix (in Finite Element Modeling). We have rewored the manuscript accordingly here to make this point clear.

- Line 324 Again, I dont think that the computational cost do not depend on n, even theoretically.Agreed, we have rephrased to convey the fact that the computation relies on one reverse sweep, hence the need to compare the computation time between one forward sweep to evaluate f, and one reverse sweep to evaluate the gradient of f.

- Figure 5-6 What is the specific application solved to produce results showed in Figures4and 5? Are these non linear problems? How many processors are you using when solving the probem with MUMPs? Please add also the overloading overhead in Figure 6, as done in Figure 5. The manuscript has been modified with respect to Figures 3 and 4 to demonstrate the applicability of the AD approach to computing the gradient of a function with respect to model inputs, as requested by the reviewer in his general remarks also. We have also added a description of the benchmark test relied upon in the text, corresponding to test 101 of the ISSM regression suite, and details about the number of cpus used with the MUMPS problem. In terms of non-linearity, test 101 is non-linear with respect to the way viscosity is treated, we have also described this further in the manuscript.

[revised manuscript text omitted]

---

## Author Response (AR2)

**1 Editor Comment #1**

The responses to the referees were thorough and thoughtful, and the changes to the paper reflect these responses. I am grateful in particular to the review from referee #1, whose understanding of the issues at hand regarding reverse-mode operator overloading is far far beyond my own. As the issues raised seem to be acknowledged and addressed, I believe the amendments are appropriate and sufficient.

I raise one small point which I would like the authors to consider or comment on before publication, and also make an additional minor suggestion. The authors are free to ignore as I will still allow publication.

– In response to referee 2 the authors have referenced Goldberg et al 2016, but i think the reason for its mention by referee 2 was not the fact that it is source-to-source but that it implements the fixed-point algorithm of christianson (1994), which does make use source-transformed code, but solves a slightly different fixed-point problem for the adjoint of the velocity solver, which is distinct from the reverse approach used here. I am not sure whether this would even be possible with OL, but outwith that I just want to flag that the source-to-source approach was not the innovation to which the referee referred.

– A very small point: I have recently learned that the canonical reference for the in-text equations at lines 236-7 is: Mike B. Giles. Advances in Automatic Differentiation, chapter Collected Matrix Derivative Results for Forward and Reverse Mode Algorithmic Differentiation, pages 3544. Springer Berlin Heidelberg, Berlin, Heidelberg, 2008

We thank the editor for his work editing this manuscript and catching issues that we had not previously dealt with. In response to comment 1, indeed the intent of our inclusion of Goldberg 2016 was to refer to the introduction of source to source AD in ice-flow models, which we have now modified to Goldberg 2013, which is more appropriate. As to comment 2, we thank the editor for this information, and we now reference Giles 2008 for equations at line 236.